# The Scores and Manner of Performing the Stand and Reach Test in Girls and Boys of Different Body Weight

**DOI:** 10.3390/bioengineering9100538

**Published:** 2022-10-09

**Authors:** Agnieszka Jankowicz-Szymańska, Justyna Kawa, Katarzyna Wódka, Eliza Smoła, Marta A. Bibro, Aneta Bac

**Affiliations:** 1Section of Physiotherapy, Faculty of Health Sciences, University of Applied Sciences in Tarnow, 33-100 Tarnow, Poland; 2SHARK Swim School, 32-002 Niepołomice, Poland; 3Institute of Applied Sciences, Faculty of Motor Rehabilitation, University of Physical Education in Krakow, 31-571 Krakow, Poland

**Keywords:** muscles’ flexibility, forward bending, movement pattern

## Abstract

Introduction: Flexibility is one of the components of Health-Related Fitness. The range of flexion has been the participant of numerous publications, but research into the quality of flexibility is lacking. The aim of the study has been to compare the scores and the quality of the stand and reach test in both overweight girls and boys and girls and boys with normal body weight. We have checked whether the forward bend movement is symmetrically distributed over the hip joints and the lumbar and thoracic spine and how it influences the position of the knee and ankle joints. Material and methods: 100 girls and 100 boys aged 10–14 years were examined. Flexibility was measured using the stand and reach test. The quality of the bend was assessed by examining the range of movement in individual body segments: the range of flexion of the thoracic and lumbar spine (linear measurements), the range of flexion of the hip joint, and the position of the knee and ankle joints at maximum flexion (angular measurements). The results were subjected to statistical analysis. Results: The participants, especially boys, had poor flexibility. A poor stand and reach test result correlated with a lower range of flexion of the thoracic and lumbar spine, greater flexion of the hip and knee joints, and greater plantar flexion at maximum torso bend position. Although the mean stand and reach score was slightly greater for the girls, gender did not significantly differentiate the way in which the stand and reach test was performed. Being overweight also did not affect the quantity or quality of the stand and reach test. Conclusions: Limitation of flexibility is common in 10–14-year-old children and results mainly from limited mobility of the spine. The compensation for this is excessive movement in the joints of the lower extremities.

## 1. Introduction

Flexibility is the ability to move freely through a full range of motion, without pain or discomfort, necessary to achieve an appropriate level of physical fitness [1]. It is considered to be one of the basic components of Health-Related Fitness, and many studies emphasize the need for stretching exercises in physical education lessons as well as a supplement to training regimens in different sports. Flexibility testing is part of any fitness assessment. However, there are no publications on the quality of flexibility tests, which consist in performing the full possible forward bend. However, it seems that the strategy for doing this exercise may be different and depends on many factors. Limitation of flexibility is believed to be a predisposing factor for musculoskeletal disorders, e.g., lumbar spine pain, injuries of the hamstrings, and even excessive muscle tension in the neck [2,3,4,5]. On the other hand, some publications disprove the relationship between the range of motion of the lumbar spine and hip joint and injuries of the hamstrings [6]. Flexibility, which is the subject of numerous scientific reports, is paradoxically a poorly understood trait of motor skills.

In assessing flexibility, the most frequently used tests are the sit and reach and the stand and reach tests (also known as the toe touch test). They are used in physical education lessons, in physiotherapy, in sports and as one of the elements of the evaluation of physical fitness of people of different ages. These tests are commonly available as they are easy and cheap to use [3,7,8]. Research shows that these tests are accurate tools, and that their scores correlate with the range of motion of the lumbar spine, hip joint and the flexibility of the hamstrings [9]. The comparison between the sit and reach and the stand and reach tests shows no significant differences, since both tests produce similar results in the assessment of flexibility. However, surface EMG reveals that the activity of the lumbar spine extensor is higher in the stand and reach test [10]. In our research, we used the stand and reach test because it seems to be more functional. The forward bend motion while standing is made multiple times each day, as opposed to the forward bending motion while sitting.

The aim of the study has been to compare the results and the quality of the stand and reach test between the girls and boys with normal body weight and those who are overweight. The objective of the research has been to discover whether gender and body weight has an influence on the amount and quality of movement. We have tried to determine whether the forward bend motion in the examined children is symmetrically distributed over the hip joints and the lumbar and thoracic spine, and how it influences the position of the knee and ankle joints. We have checked whether any body segments are hypermobile and require stabilization exercises and hypomobile and require flexibility.

## 2. Materials and Methods

### 2.1. Characteristics of the Studied Group

Two hundred students were invited to take part in the study, including 100 girls and 100 boys, aged from 10 to 14 from a primary school in a small town (106,000 inhabitants) in southern Poland (the number of girls and boys from each age group is given in Table 1). All students and their parents accepted the invitation and gave their written consent to participate in the research. The school principal also consented to the study. Participant recruitment and research were carried out at the beginning of 2020. Patient names and surnames were coded, and the authors did not have access to them after the data had been collected.

The inclusion criteria were as follows: no chronic diseases, no congenital or acquired malformations of the musculoskeletal system (data from the parents), no certificate of disability (data from the school nurse) and feeling well on the day of the study (information from the child just before the start of the measurement). Sexual maturity and body proportions were not assessed in the girls and boys, as studies indicate no significant influence of these features on the results of the flexibility test [2,11,12]. According to the interviews, the girls and boys participating in the study did not train regularly in any sports club. Their physical activity was limited to compulsory PE lessons at school and spontaneous sports activities (walking, cycling, etc.).

The research was conducted in accordance with all the guidelines of the Helsinki Declaration. All tests and measurements were performed in the nurse’s office by the same experienced person, using the same research tools. When made readable, the results were read to another person who entered them in a table. The range of hip flexion and the position of the knee and ankle joint in the bend were always read on the left side of the participant’s body. The measurements were carried out in the afternoon. The participants were dressed in sports clothes, a T-shirt and shorts, and they were not wearing shoes. 

### 2.2. Research Procedures

#### 2.2.1. Determination of the Body Weight Status

Body height and weight were the basis for calculating the BMI. Body height was tested with a calibrated anthropometer (Alumet, Warsaw, Poland) from the Basis point to the Vertex point with an accuracy of 1 mm. Body weight was tested on a TANITA electronic scale (Tanita Corporation of America Inc, Arlington Heights, IL, USA) with an accuracy of 0.1 kg. To determine the body weight, we relied on the findings of Cole et al. [13] in which the authors propose the BMI threshold overweight and obesity values for girls and boys at a given age.

#### 2.2.2. Stand and Reach Test 

The first stage before carrying out the test was to mark four points on the child’s spine with a marker, which were later used for the Otto test and the Schober test (see the description below). Then, the participant was asked to stand on a special step with a vertical measuring bar, on which the ‘0′ point was at the height of the surface on which the participant was standing, the positive values increased towards the ground to 25 cm, and the negative values increased going upwards to −35 cm (Figure 1). The feet were together, and the toes were aligned with the edge of the step. The participant made a free bend forward and was asked to keep their legs as straight as possible in the knee joints, the arms hanging down and straight, and the head as straight as possible. The knee joints were not stabilized on purpose to check if the participants would automatically change their position in the bend. In the position of the maximum forward reach, the examiner read the result of the stand and reach bend, i.e., the value on the measuring bar, which the participant was reaching with the tip of their longest finger. Subsequent measurements and tests were performed in the position described above (Figure 1). 

#### 2.2.3. Otto Test

The Otto test measures the extent to which the thoracic spine can flex.

In order to perform the test, before the participant began the bend, in a standing position, the spinous process C7 was marked with a marker, and then 30 cm down the spine, another point was marked. In the position of the bend, the distance between the previously marked points was measured again and it was calculated if the distance was greater than 30 cm. The normal difference should be between 2 and 4 cm [14].

#### 2.2.4. Schober Test 

The Schober test determines the range of the lumbar spine flexion. In order to perform it, before starting the bend, in a standing position, the spinous process S1 was marked with a marker, and then the next point was marked 10 cm up the spine. In the position of the bend, the distance between the previously marked points was measured again and it was calculated whether the distance was greater than 10 cm. The normal difference should be 5 cm [14].

#### 2.2.5. The Range of Hip Flexion

A goniometer was used to determine the hip flexion angle in the bend position. The axis of the goniometer was aligned with the transverse axis of the joint on the greater trochanter of the femur. The movable arm was pointing at the crest of the iliac ala, while the stationary arm was directed towards the arrow’s head. The result was read in degrees [15].

#### 2.2.6. The Position of the Knee Joint

In order to determine the position of the knee joint in a bend, the axis of the goniometer was positioned near the arrowhead, parallel to the transverse axis of the joint. The movable arm of the goniometer was pointing towards the greater trochanter of the femur, while the movable arm-towards the lateral malleolus. The result was read in degrees. When the knee joint was in a neutral position (neither flexion nor hyperextension), the goniometer showed 180°. Lower values meant flexion, higher values-hyperextension.

#### 2.2.7. The Position of the Ankle Joint

In order to determine the position of the ankle joint in the bend position, the axis of the goniometer was placed below the ankle. The stationary arm was positioned along the 5th metatarsal bone on the outer edge of the foot, parallel to the ground, while the movable arm was projected at the arrowhead. The result was read in degrees.

### 2.3. Statistical Analysis

The collected data was analyzed using the Statistica 13 software. Basic descriptive statistics and frequency tables were used. The Shapiro–Wilk test was used to ascertain whether the distribution of variables is normal, and the F test to test the homogeneity of variance. If the compared independent variables had a normal distribution and uniform variances, the differences between groups were tested using Student’s *t*-test for independent samples. When the distribution of variables deviated from the norm, the Mann–Whitney U test was used. The relationships between the variables were investigated using Pearson’s linear correlation. The level of significance was set at α = 0.05. 

## 3. Results

The examined girls and boys did not differ significantly in terms of body weight, height and BMI (Table 2). Correct body weight was diagnosed in 158 (79%) of 200 children, including 78 boys and 80 girls. Twenty-two boys and twenty girls were overweight. Obesity was not found in any of the studied children.

The overweight and normal-weight children did not differ either in the stand and reach test scores (which were slightly better in overweight children), or in the range of flexion of the thoracic (Otto test) and lumbar (Schober test) spine. The position of the hip and ankle joints in the forward bend also did not differentiate between the groups. The only examined variable that significantly distinguished overweight children and children with normal body weight was the position of the knee joint in the bend. Overweight children’s knee joint was closer to the fully extended position (180°), while children with normal body weight had greater flexion in the knee joint. The difference was 2.72° (Table 3).

The girls had slightly better stand and reach test results (the difference was 1.52 cm). During the forward bend, girls tended to use the flexion of the thoracic spine to a lesser extent (the difference of 0.12 cm), and the flexion of the lumbar spine to a greater extent (the difference of 0.32 cm). In the girls’ full bend, the hip and knee joints were less flexed (difference 1.14° and 0.19°), and the ankle joint had greater plantar flexion (difference 1.65 °) but the differences were not statistically significant (Table 3).

In the examined children, the quality of flexibility was determined on the basis of the stand and reach test scores. It was assumed that reaching the surface of the step on which the child was standing with the tip of their longest finger (value ‘0′ on the measuring bar) means good flexibility, while poor flexibility was diagnosed when the child was unable to reach this point.

One in four children was diagnosed with good flexibility, of whom girls were found to have good flexibility slightly more frequently. Among 158 children with normal body weight, a good stand and reach test result was recorded in 24% of the participants, while among 42 overweight children, a good stand and reach test result was found in 31% of the participants (Table 4).

Children with normal and poor flexibility did not differ in height, weight, or BMI. In children with normal flexibility, a greater range of flexion of the thoracic and lumbar spine, less flexion of the hip and knee joints and less plantar flexion of the ankle joint were observed in the position of a full forward bend (Table 5).

Significant, though weak, correlations were found between the stand and reach test results and the mobility of the thoracic and lumbar spine as well as the position of the hip, knee and ankle joint in the bend. The analysis of the same correlations across gender groups and body weight status confirmed these observations: a better stand and reach test score was associated with a greater range of movement of the thoracic and lumbar spine (positive correlations) and less flexion of the hip joint in the bend position (negative correlation) (Table 6).

## 4. Discussion

Girls and boys aged between 10 and 14 had poor flexibility. It was found that the assessment of the stand and reach helped identify children with limited flexibility, but additional tests were necessary to assess the quality of the bend. Our research showed significant differences in the method of performing the bend by children with good and poor flexibility. Girls and boys with a good flexibility test score used more thoracic and lumbar spine flexion and held the knee and ankle joints closer to the neutral position. In children with limited flexibility, reduced flexion within the spine joints, greater flexion of the hip and knee joints and greater plantar flexion of the ankle joint were noted. Moreover, it was found that gender and also the body weight status had not an influence on the result of the stand and reach test and the quality of the performance of the bend.

There was no evidence of an influence of being overweight on the result and quality of the stand and reach test. Nikoladis’s research also indicates the lack of a relationship between BMI and body fat and flexibility in adolescents and adults [16]. On the other hand, Hands et al. [17], different to our study, diagnosed better results of flexibility in teenage girls compared to boys of the same age.

The average result of the Schober test indicated limited mobility of the lumbar spine in the examined children. This finding was true for both genders, with the range of lumbar spine motion being slightly smaller in boys. 

According to Kendall et al. [18], the correct range of hip motion in the forward bend (sit and reach) is around 80° (the angle between the sacrum and the horizontal line). Comerford and Mottram [19] determined the correct range of hip flexion during a forward bend while standing at 70°. Comparing our findings with the conclusions of the above-mentioned authors, we observed that the examined children used the hip joint excessively. At the same time, the participants found it difficult to extend the knee joint. The question is: does this tendency indicate a weakening of the hamstring muscles in the upper segments and the shortening of these muscles in the lower segments? Perhaps this hypothesis can be explained by the sedentary lifestyle that is very common today. Of course, our research does not answer this question, but only motivates the search for the answer. Rakholiya et al. [20] indicated in their research that a sedentary lifestyle leads to an imbalance of hip extensor. On the other hand, the influence of the sedentary life on the flexibility of hamstrings is unclear. One study shows that long-lasting sitting can lead to hamstring tightness [21], others contradict this thesis [22]. However, the above-mentioned studies investigated the elasticity of hamstrings as a whole, without taking into account that their proximal and distal segments, acting on a different joint (as hip extensors or knee flexors), may have different flexibility.

The stand and reach and sit and reach tests are often used to evaluate the flexibility of the hamstrings. According to Magnusson et al. [23], the range of the forward bend is a good measure of the flexibility of these muscles. As Chillon et al. [24] state, the hip angle explains 42% of the sit and reach test result, the lumbar angle-30% and the thoracic angle-just 4%. The research carried out by Muyor et al. [8] shows, however, that the sit and reach test correlates poorly or at most moderately with the flexibility of the hamstrings as assessed by the passive strait leg raise test. It must be remembered that both the sit and reach test and the stand and reach test are indirect measures and the limitation of flexibility diagnosed by one of these tests cannot be a direct indication for stretching the hamstrings [8]. Stretching the structures that are not shortened leads to hypermobility which, like hypomobility, can cause injury and reduced sport performance [25].

### 4.1. Limitations

It seems that in further studies on flexibility, the flexibility of the hamstrings and the flexibility of the calf muscles should be additionally assessed in an isolated study.

### 4.2. Clinical Implications

It was our intention that our findings should stimulate further research to seek answers to the following questions: What is correct flexibility? Is the range or the quality of the movement more important? How to find a compromise between these features? Should the approach to stretching exercises change: first the analysis of the quantity and quality of the forward bend, then individually selected exercises for segmental increase in flexibility and stabilization of hypermobile segments?

## 5. Conclusions

Children aged from 10 to 14 have poor flexibility.Girls achieve similar results on the flexibility test (stand and reach) than boys at the same age.Being overweight affects neither the quantity nor the quality of the stand and reach bend. The only significant difference in the way the bend is done by overweight children is the more correct (closer to neutral) position of the knee joint.Participants who score poorly in the stand and reach test have a smaller range of flexion of the thoracic and lumbar spine. When performing the bend, they use the flexion of the hip and knee joints to a greater extent and place the ankle joint in the position of greater plantar flexion.

## Figures and Tables

**Figure 1 bioengineering-09-00538-f001:**
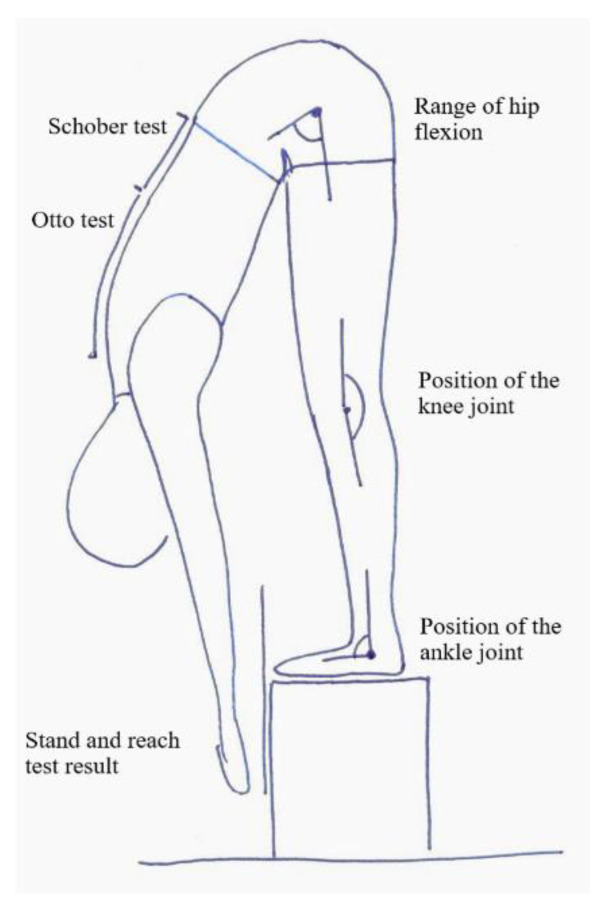
Measurement scheme.

**Table 1 bioengineering-09-00538-t001:** The number of girls and boys broken down into age groups.

Age [Years]	Boys	Girls	All
10	26	12	38
11	19	17	36
12	19	21	40
13	18	27	45
14	18	23	41
All	100	100	200

**Table 2 bioengineering-09-00538-t002:** Comparison of the physique of girls and boys (differences were considered significant for *p* < 0.05).

Variable	Group	Mean	Min.	Max.	St. Dev.	*p*
Body weight [kg]	Boys	48.16	32.80	65.00	7.82	0.955
	Girls	48.56	35.00	68.00	6.65	
Body height [cm]	Boys	153.95	132.00	175.50	11.82	0.631
	Girls	153.39	132.00	175.00	10.01	
BMI [kg/m^2^]	Boys	20.24	16.98	24.34	1.25	0.101
	Girls	20.57	18.67	24.38	1.24	

Min.—Minimum; Max.—Maximum; St. Dev.—Standard Deviation.

**Table 3 bioengineering-09-00538-t003:** Comparison of the position of the individual segments of the legs and the spine in the forward bend.

Body Weight	Mean	Min.	Max.	St. Dev.	*p*	Variable	Gender	Mean	Min.	Max.	St. Dev.	*p*
Overweight	−3.17	−20.00	12.00	5.99	0.150	Stand and reach test [cm]	Boys	−5.13	−28.00	10.00	7.53	0.346
Normal	−4.69	−28.00	15.00	7.87			Girls	−3.61	−28.00	15.00	7.49	
Overweight	3.76	1.50	5.00	1.00	0.350	Otto test [cm]	Boys	3.67	0.50	6.50	1.13	0.577
Normal	3.57	0.00	6.50	1.20			Girls	3.55	0.00	5.00	1.20	
Overweight	3.58	0.00	7.00	1.43	0.320	Schober test [cm]	Boys	3.57	0.00	7.00	1.43	0.088
Normal	3.77	0.00	7.00	1.30			Girls	3.89	1.20	7.00	1.19	
Overweight	115.76	80.00	140.00	11.34	0.741	Hip joint position [°]	Boys	117.19	80.00	150.00	11.40	0.119
Normal	116.85	80.00	150.00	9.55			Girls	116.05	85.00	130.00	8.23	
Overweight	177.57	165.00	200.00	6.50	0.010 *	Knee joint position [°]	Boys	175.33	160.00	190	5.96	0.986
Normal	174.85	160.00	190.00	5.59			Girls	175.52	164	200	5.83	
Overweight	102.50	80.00	125.00	9.89	0.576	Ankle joint position [°]	Boys	102.18	70.00	125.00	9.09	0.245
Normal	103.14	70.00	125.00	8.13			Girls	103.83	85.00	125.00	7.85	

Min.—Minimum; Max.—Maximum; St. Dev.—Standard Deviation; *—statistically significant differences.

**Table 4 bioengineering-09-00538-t004:** The stand and reach test result in qualitative assessment.

Stand and Reach Test-Interpretation	Gender	Body Weight Status Normal	Body Weight Status Overweight	Total in Row
Normal flexibility-the participant reached at least the feet support surface with their fingertips	Boys	16	7	23
		69.57%	30.43%	23.00%
	Girls	22	6	28
		78.57%	21.43%	28.00%
	All	38	13	51
		74.51%	25.49%	
Low flexibility-the participant did not reach the feet support surface with their fingertips	Boys	62	15	77
		80.52%	19.48%	77.00%
	Girls	58	14	72
		80.56%	19.44%	72.00%
	All	120	29	149
		80.54%	19.46%	
Total in column		158	42	200

**Table 5 bioengineering-09-00538-t005:** Variable values based on good or poor flexibility.

Variable	Stand and Reach Test-Interpretation	Mean	Min.	Max.	St. Dev.	*p*
Body weight [kg]	Normal flexibility	48.67	34.00	62.00	6.85	0.511
	Low flexibility	48.26	32.80	68.00	7.40	
Body height [cm]	Normal flexibility	153.83	132.00	173.00	11.12	0.912
	Low flexibility	153.61	132.00	175.50	10.90	
BMI [kg/m^2^]	Normal flexibility	20.54	18.12	24.34	1.33	0.516
	Low flexibility	2036	16.98	24.38	1.23	
Otto test [cm]	Normal flexibility	3.94	0	5.00	1.26	0.002 *
	Low flexibility	3.49	0	6.50	1.11	
Schober test [cm]	Normal flexibility	4.67	3.00	7.00	0.89	0.001 *
	Low flexibility	3.41	0	7.00	1.30	
Hip joint position [°]	Normal flexibility	111.92	80.00	125.00	9.85	<0.001 *
	Low flexibility	118.23	80.00	150.00	9.47	
Knee joint position [°]	Normal flexibility	177.68	165.00	200.00	5.65	0.013 *
	Low flexibility	174.65	160.00	190.00	5.77	
Ankle joint position [°]	Normal flexibility	98.92	80.00	120.00	8.00	<0.001 *
	Low flexibility	104.40	70.00	125.00	8.25	

Min.—Minimum; Max.—Maximum; St. Dev.—Standard Deviation; *—statistically significant differences.

**Table 6 bioengineering-09-00538-t006:** Correlations between the stand and reach test results and the studied variables in groups distinguished by gender and body weight status.

Group	Body Weight [kg]	Body Height [cm]	BMI [kg/m^2^]	Otto Test [°]	Schober test [°]	Hip Joint Position [°]	Knee Joint Position [°]	Ankle Joint Position [°]
Boys	−0.107	−0.151	0.118	0.200 *	0.189	−0.298 *	0.120	−0.174
Girls	0.018	0.050	−0.035	0.326 *	0.275 *	−0.243 *	0.236 *	−0.195
Overweight	−0.155	−0.159	−0.017	0.147	0.327 *	−0.259	0.230	−0.280
Normal body weight	−0.014	−0.014	0.017	0.271 *	0.228 *	−0.282 *	0.154	−0.147
All	−0.047	−0.061	0.055	0.257 *	0.237 *	−0.276 *	0.178 *	−0.172 *

*—statistically significant correlations.

## Data Availability

The datasets used and/or analysed during the current study available from the corresponding author on reasonable request.

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
