# Peer review of "The Scores and Manner of Performing the Stand and Reach Test in Girls and Boys of Different Body Weight"

_bioengineering, 2022, doi:10.3390/bioengineering9100538_

Round 1

Reviewer 1 Report

Itis not clears from the experimental procedures if the people sample has been chosen among boys and girls with previous experiences in soprt activities. this point should be clearly specified (interviews were carried out and a homogeneous sample built?)
The legenda of figure 1 the Authors should specify the name and site of the measurements (also indicating id lenght or angle), nothing should be delegate to the Reader interpretation. Since it is hyprthesised that the observed boys and girls low flexibility could be related to the sedentary life style of the population investigated, reference values of the measured values for categories such trained in differerent sports young adolescents.

Whithout a reference the study is not significant.

Author Response

On behalf of all the authors, I would like to thank you for reviewing our manuscript. Thank you for your comments. We have made changes and hope that these will meet the expectations of both reviewers.

Review 1

Itis not clears from the experimental procedures if the people sample has been chosen among boys and girls with previous experiences in soprt activities. this point should be clearly specified (interviews were carried out and a homogeneous sample built?)

Thank you for this comment. We did indeed overlook an important piece of information. This has now been completed.

Lines 78-80: According to the interviews, the girls and boys participating in the study did not train regularly in any sports club. Their physical activity was limited to compulsory PE lessons at school and spontaneous sports activities (walking, cycling etc.).

The legenda of figure 1 the Authors should specify the name and site of the measurements (also indicating id lenght or angle), nothing should be delegate to the Reader interpretation.

Descriptions have been added to Figure 1.

Since it is hyprthesised that the observed boys and girls low flexibility could be related to the sedentary life style of the population investigated, reference values of the measured values for categories such trained in differerent sports young adolescents. Whithout a reference the study is not significant.

Thank you very much, I agree that this needed to be completed. It has been corrected on lines 231-239.

English is not our native language, so when preparing the manuscript, we commissioned a professional translation agency that employs native speakers to proofread the manuscript. The text of certificate is presented below.

I hereby declare that the article entitled ‘Do gender and body weight status differentiate the scores and manner of performing the stand and reach test?’ by Agnieszka Jankowicz-SzymaÅ„ska, Justyna Kawa, Katarzyna Wódka, Eliza SmoÅ‚a, Marta Bibro and Aneta Bac has been proofread by a native speaker of English, a professional proofreader of academic articles.
Yours sincerely,
Barbara Komorowska
Translation Agency UZUS
Biuro Tłumaczeń UZUS
ul. Winogrady 120 A
61-626 Poznań
http://www.uzus.pl
e-mail: [email protected]
tel. (+48) 606 630 989

Reviewer 2 Report

General Comment

     Flexibility is considered one of the physical fitness, and lower flexibility is associated feature back pain risk. In this study, the authors compared the quality of flexibility tests between normal and overweight boys and girls. According to the results, they did not find differences in flexibility between normal and overweight. However, silently gender difference was found. Those results may be useful for scoring physical fitness tests in school children.

Major Comment

Overweight children likely have lower flexibility. Unfortunately, those results did not support the hypothesis. However, not only positive results but also negative results could advance the understanding of physical fitness, such as flexibility in children.

     If possible, please add the other physical fitness level such as sprint, endurance, and power score, daily physical activity levels, and sports history. It may influence flexibility. The overweight groups are small sample sizes (22 boys and 20 girls), so those characteristics strongly affect the results. Authors carefully evaluated the quality of flexibility, so other specific data improves reliance and valuable the manuscript.

Author Response

On behalf of all the authors, I would like to thank you for reviewing our manuscript. Thank you for your comments. We have made changes and hope that these will meet the expectations of both reviewers.

Review 2

General Comment

     Flexibility is considered one of the physical fitness, and lower flexibility is associated feature back pain risk. In this study, the authors compared the quality of flexibility tests between normal and overweight boys and girls. According to the results, they did not find differences in flexibility between normal and overweight. However, silently gender difference was found. Those results may be useful for scoring physical fitness tests in school children.

Major Comment

Overweight children likely have lower flexibility. Unfortunately, those results did not support the hypothesis. However, not only positive results but also negative results could advance the understanding of physical fitness, such as flexibility in children.

Thank you for this comment.

     If possible, please add the other physical fitness level such as sprint, endurance, and power score, daily physical activity levels, and sports history. It may influence flexibility. The overweight groups are small sample sizes (22 boys and 20 girls), so those characteristics strongly affect the results. Authors carefully evaluated the quality of flexibility, so other specific data improves reliance and valuable the manuscript.

We are sorry, but we cannot complete the manuscript with the results of other physical fitness tests as we did not do them in this project.

English is not our native language, so when preparing the manuscript, we commissioned a professional translation agency that employs native speakers to proofread the manuscript. The certificate is presented below.

I hereby declare that the article entitled ‘Do gender and body weight status differentiate the scores and manner of performing the stand and reach test?’ by Agnieszka Jankowicz-SzymaÅ„ska, Justyna Kawa, Katarzyna Wódka, Eliza SmoÅ‚a, Marta Bibro and Aneta Bac has been proofread by a native speaker of English, a professional proofreader of academic articles.
Yours sincerely,
Barbara Komorowska
Translation Agency UZUS
Biuro Tłumaczeń UZUS
ul. Winogrady 120 A
61-626 Poznań
http://www.uzus.pl
e-mail: [email protected]
tel. (+48) 606 630 989

Round 2

Reviewer 1 Report

The paper has been improved by the required modifications

Reviewer 2 Report

The authors revised the manuscript appropriately.